# Educational Intervention in the Postural Hygiene of School-Age Children

**DOI:** 10.3390/healthcare10050864

**Published:** 2022-05-06

**Authors:** María José Menor-Rodríguez, Raquel Rodríguez-Blanque, María Montiel-Troya, Jonathan Cortés-Martín, María José Aguilar-Cordero, Juan Carlos Sánchez-García

**Affiliations:** 1Research Group CTS-367, Ourense Nursing Department, University of Vigo, 36310 Vigo, Spain; mariajosemenor@hotmail.com; 2Research Group CTS-1068, Nursing Department, University of Granada, 18011 Granada, Spain; rarobladoc@ugr.es (R.R.-B.); jonathan.cortes.martin@gmail.com (J.C.-M.); mariajaguilar@telefonica.net (M.J.A.-C.); jsangar@ugr.es (J.C.S.-G.); 3Research Group CTS-1068, Nursing Department in Ceuta, University of Granada, 51001 Ceuta, Spain

**Keywords:** educational intervention, children, postural hygiene, children’s health

## Abstract

Introduction: Healthy lifestyle habits formed in childhood provide the foundation for a healthy adult life; therefore, it is important to encourage healthy habits and to correct poor habits from an early age. Aim: In this study, we aim to evaluate the influence of educational intervention in order to modify postural hygiene habits in school-age children. Materials and methods: We randomly selected three public primary education schools in the Galician provinces of Orense and Pontevedra based on stratified multistage sampling. A sample of 479 students was obtained, representing 2% of all 6- to 12-year-old children registered during the 2015/2016 academic year in those provinces. Results: Following the intervention, the students’ postural hygiene improved. We found statistically significant differences regarding a reduction in the number of hours spent watching television (0.531 h/day); the way school books and supplies were carried, with an increase in the use of backpacks with wheels (from 58.5% to 64.1%); and an improvement in postural hygiene when watching television, with an increase from 63.7% to 80.8% of those surveyed opting to watch television whilst seated on a chair instead of lying down. Conclusions: Educational intervention by inculcating healthy postural hygiene habits in children at an early age can improve and correct unhealthy behaviours.

## 1. Introduction

Healthy lifestyle habits are an essential component of disease prevention, particularly those that are directly related to negative conducts that affect our health and those that can lead to deaths worldwide. These conducts depend on the behaviour and attitude of people in their daily lives, as good health depends to a large degree on diet and lifestyle [1,2,3].

Many diseases significantly affect the quality of life of those affected, so healthy lifestyle habits and disease prevention tend to be inextricably linked [4,5,6].

Promoting health is one strategy for acquiring and developing personal skills relating to behavioural changes regarding health and encouraging a healthy lifestyle [7,8].

The WHO believes that schools play a key role in developing healthy lifestyle habits. The relationship between educational centres and health is unquestionably helpful in correcting habits that are unhealthy and has an effect on the continuance of newly acquired habits with a view to the future. Implementing health-related educational programmes in schools is essential for ensuring maximum dissemination and bringing about favourable results in terms of change [9].

The scientific community is currently focusing on devising ways to promote a healthy lifestyle during school years, and the scientific literature presents a wide range of resources for this purpose. There seems to be a consensus on educational intervention as an effective resource during the school years when the effect of peer group influence is a factor to consider in the acquisition of new habits [8,10,11]. Additionally, it is more likely that children will maintain those adequate lifestyle habits as adults, continuing to practise sports, with regular physical exercise and following a healthy diet [12,13].

Several international authors have highlighted the fact that working on aspects related to actions of universal prevention for the promotion of a better quality of life in school settings is of great value, as it is more attainable and effective to develop and maintain healthy behaviours in younger children than to try and change unhealthy habits in adulthood [14,15,16].

Leixà (2003) has proposed that education for health is a process of information and individual responsibility in order to acquire knowledge, attitudes and basic habits for the protection and promotion of individual and collective health [17].

There are many international proposals backed by important organisations such as the Pan-American Health Organisation (PHO), the World Health Organization (WHO), UNESCO (United Nations Education, Science and Culture Organisation) and the FAO (Food and Agriculture Organisation for the United Nations) that are beginning to recognise good practices in the promotion of health in school settings, many of them in alliance with the business sector [18].

One health problem that has arisen with our new modern sedentary lifestyle is commonly known as non-specific back pain, defined as the pain that appears in the lumbosacral spine and which frequently leads to restricted movement. Currently, back pain is a common public health issue [19], and it has also been described as a public health problem in children and teenagers [20].

A preventative approach helps diminish back pain. Prevention can be achieved through the acquisition of knowledge and/or an improvement of correct postural habits, both of which favour back care in children and teenagers [21].

Outside school settings, the concept of “back schools” was created for the general population with the aim of teaching postural education. The programme includes information on the anatomy and function of the spine, mechanisms that cause pain, pain management, correct posture, techniques for lifting and carrying heavy loads, and stretching and toning exercises [22].

“Back schools” act on two levels [23]: primary prevention and secondary prevention. Secondary prevention treats people with spinal disorders with the aim of improving their condition. Primary prevention is aimed at healthy people to teach them behaviours that will allow them to care for their spine in all of their everyday activities. This is the most relevant level in school settings.

Innovative techniques have been developed based on participatory teaching methods that use body language and expression as a means of transmitting information on the most common everyday postures (sitting, picking up objects, carrying backpacks, etc.) with the aim of involving students in their own learning process. To do so, simple role plays and sketches are designed by the students in small groups, following flexible instructions, through which they teach their peers the key aspects of correct posture. This innovative method gives the lessons greater meaning, and what is learnt can be transferred to their daily activities, leading to improvements in their quality of life [24,25,26,27].

We have taken into account health problems in the child population and previous references that highlight the importance of educational intervention to promote healthy lifestyles. This work raises the following scientific question: How does educational intervention influence schoolchildren to promote healthy postural hygiene habits?

Aim:

In this study, we aim to evaluate the influence of an educational intervention carried out over a period of 28 weeks on modifying postural hygiene habits in school-age children.

## 2. Materials and Methods

### 2.1. Design

A pre-post intervention study was designed as an educational intervention to improve postural hygiene in school children. It was approved by the Research Ethics Committee for the province of Granada, assigned File No. 13-06 and registered with the registration code 2014/130 in the CEI of Pontevedra-Vigo-Ourense (https://acis.sergas.gal/cartafol/Axentes-de-investigacion accessed on 1 January 2022). At all times, the study was conducted in accordance with the provisions of the Declaration of Helsinki, as amended at the 64th WMA General Assembly, Fortaleza, Brazil, October 2013.

Informed consent was obtained from all subjects involved in the study.

### 2.2. Participants

The study population was the total number of children registered in all Primary Education schools in the Galician provinces of Orense, both in rural and urban settings, and Pontevedra, specifically in the city of Vigo, during the academic year 2015/2016, n = 24,900.

From the total population, we performed a randomised two-phase sampling to select the participant schools:

In the first phase, we performed a simple random sampling to select schools from a nominal list provided by the Education Board (Delegación de Educación).

The second phase consisted of a cluster sampling that included all of the students of the previously selected schools who were in the school years chosen for the study.

### 2.3. Sample Size

We selected a sample of 479 students, an equivalent of 2% of all children aged 6 to 12 registered in the public schools of the provinces of Orense and Pontevedra during the academic school year 2015/2016.

The required sample size of n = 479 students was calculated by establishing a confidence level of 95% and a margin of error of 5%, considering that the total number of students aged 6–12 years registered in the public schools of the provinces of Orense and Pontevedra was 24,900 students and assuming that, after the educational intervention, in the worst case scenario, only 50% of the students would show improvements.

### 2.4. Inclusion Criteria

Students were between the ages of 6 and 12 years from the three selected schools. The students were required to have read the informed consent, alongside their parents or legal guardians, who were responsible for signing it if they agreed to their child participating in the intervention.

### 2.5. Exclusion Criteria

Existing physical or psychological limitations that prevented the student from understanding the questions and answers of the questionnaire, and refusal to respond to the survey freely and voluntarily were the exclusion criteria.

### 2.6. Educational Intervention

The educational intervention was carried out in three phases, as was the data collection process.

Initial evaluation: we evaluated postural hygiene habits using a questionnaire containing questions from the program PERSEO [28] and the ENKID study [29], which were adapted with pictograms to facilitate comprehension in the case of younger children. We also collected information on the family and sociodemographic context of the students participating in the study, assessing aspects such as the age of the parents and their employment status, number of siblings and the people with whom they lived on a daily basis.

Educational intervention: we implemented the previously designed educational intervention through formative activities on postural hygiene adapted for the different age groups of the children participating. The duration of the intervention was one academic school year (28 weeks), and it was aimed at motivating and involving the students in the process of improving their postural hygiene by using educational videos, workshops and practical exercises. At all times, the intervention was proposed as a participatory game both in the school environment and at home. By means of this measure, the study attempted to gauge the active involvement of students and their families.

The activities were held outside classroom hours, following recommendations from the schools, at the end of the morning and prior to the midday break. Extracurricular activities were never carried out to avoid teaching overload.

During the intervention, follow up of students remained ongoing with tutorials.

All of the educational activities were mainly participatory, including animation techniques to create a family atmosphere, classes with a simple and practical message, group discussions and demonstration sessions.

Final evaluation: Seven months after initiating the educational intervention, we measured its effectiveness by using the same scores as those of the initial evaluation. In recognition of their efforts, all of the participants were awarded with a diploma signed by the authors of the study.

The flow diagram below represents the sample selection (Figure 1).

### 2.7. Study Variables

We analysed the following variables:

Sociodemographic variables: age, sex (boy/girl), number of siblings, the school location (urban coastal/urban inland/rural) and members of the family unit;

Postural habits related to the usual means of carrying school books and materials: backpack, backpack with wheels or trolley, or by hand;

Posture when watching television and/or playing on the computer: sitting in a seat with backrest, sitting in a seat without a backrest or lying down.

### 2.8. Statistical Analysis

We performed a descriptive analysis of the characteristics of the population studied. The quantitative variables were calculated using measures of central tendency (mean and median), dispersion (standard deviation) and position (limits of the distribution). The qualitative variables were expressed using absolute frequencies and relative frequencies (percentages).

We performed the Kolmogorov–Smirnov test to confirm the normal distribution of the variables. The variables that did not meet the normality criteria were studied using non-parametric tests.

The Chi-Square Test was used to compare the proportions of more than two independent groups. In the comparison of the pre- and post-intervention groups, the Paired T-Student Test was used for variables with normal distribution, and the Wilcoxon Test was used for variables with asymmetric distribution. McNemar’s test was conducted to examine whether there was any difference in the proportions.

We set an α error of 5% (*p* ≤ 0.05). Statistical analysis was performed using the programme IBM SPSS version 22.

## 3. Results

### 3.1. Description of the Participants

Of the 24,900 students aged 6 to 12 years registered in the public schools in the Spanish region of Galicia during the school year 2014–2015, 479 participated in the programme and all finished the study (221 boys and 258 girls). The baseline characteristics of the population that participated in the project are shown in Table 1.

### 3.2. Postural Habits Pre- and Post-Intervention

The analysis of the postural habits and their association with family structure, and the changes that occurred can be observed in Figure 2.

The mean time spent watching television was 2.53 ± 0.71 h per day. Although the number of hours per day spent watching television decreased (2.29 ± 0.531 h/day), this was not significant (*p* = 0.531).

We found statistically significant differences in the way school books and materials were carried and ub family structure (*p* < 0.001), with wheeled backpacks being the choice in families with both parents (63.1%) and in those where parents and grandparents were present (63.4%).

There was a significant change concerning postural hygiene (*p* < 0.001) and the means of carrying school materials (*p* < 0.001). The use of backpacks with wheels increased after the educational intervention from 58.5% to 63.8% (Figure 3).

## 4. Discussion

The relationship between schools and health is instrumental in correcting bad health habits, as shown in the ALADINO study [9].

The age of the students participating in the educational programme is an important factor. The population targeted by the intervention in this study comprises students aged between 6 and 12 years, which is the ideal age to promote and introduce healthy habits in terms of postural hygiene. Had this intervention been implemented in another age group, it would have obtained less favourable results. In the study by Pérez et al. on health promotion through educational programmes, age was confirmed to be a factor to be taken into account, since all programmes implemented at ages between 6 and 12 years obtained positive results [30].

Educational programmes must have a clear objective and an updated and attractive development plan for their target population [30]. In this case, it was decided to orient the interventions using techniques to capture the attention of the students without requiring any added effort, through games, participatory talks and video viewing [31].

Currently, school-age children prefer to spend their leisure time and entertainment playing video games, watching TV or using social networks, all of which implie a considerable increase in the rate of sedentary lifestyle, as confirmed by the Aladdin study [32]. In any case, all of these activities involve spending a lot of time in front of a screen, and it is necessary to pay attention to posture during this screen time in order to ensure a healthy lifestyle [33].

As the results of this study state, the posture adopted by schoolchildren in front of the television depended on the family structure to which they belonged. This fact has been confirmed by other studies such as that of Perea et al. [34].

The three most common postures adopted during school years are those of attention; usually in school classes and extracurricular activities, and writing; when taking notes and doing homework; and the rest [35,36,37]. All of these positions must be observed; analysed; and if necessary, corrected so that there will be no pathological consequences in future. Repeated exposure to an activity carried out with poor posture will cause back injuries, bone deformities, pain and defects in the human figure with aesthetic implications, affecting other physiological processes of the organism such as breathing, digestion and blood circulation [30,38].

Another aspect considered in this study was the transport of school materials. The relationship between posture while transporting materials and the incidence of spinal pathologies has also been studied by Garrido Martinez de Salazar et al., confirming that postural hygiene when carrying a backpack is essential to prevent back pain in school-age children [30].

One of the main reasons that school children aged between 6 and 12 years consult a doctor is for back pain, and this further justifies the existence of educational programmes that promote healthy lifestyle habits related to postural hygiene [31].

We envisage a need to expand our study in order to take into account variables such as a student’s back pain pre- and post-intervention in order to establish an association between back pain and the weight of their backpacks. In future, it would also be of value to draw up a set of recommendations for correct posture and practical exercises that would minimise development of spinal disorders in adulthood.

In our study, the most frequent posture when watching television was sitting in seats with backrests, although a considerable number of children (29%) did so in seats without backrests and 8% were lying down. We also found that school books and educational materials were not correctly transported, as nearly 40% of the students did not use backpacks with wheels. These results are similar to those found in other studies such as that carried out by the Sociedad Española de Pediatría de Andalucía Occidental y Extremadura [9], which studied common habits when carrying school backpacks and their relation to back pain, or the study by Garrido Martínez de Salazar et al. [30], which considered the habits and ways of carrying backpacks obtained through a survey answered by the parents of 588 school children from six schools in the Cádiz Bay area and found that 64% of students carried their backpacks on their backs.

This study reinforces the strategic role of schools in improving health-related behaviours. It would be of interest to systematically establish educational programmes with physiotherapy and sports science professionals in schools in order to correct inadequate posture and to minimise the weight of backpacks. Similar conclusions were reached by researchers such as Gallardo Vidal et al. [31], who in their article “Evaluation of the effectiveness of an educational intervention to decrease school backpack weight in 3rd and 4th year primary school children” stated that an educational intervention that reviewed the school materials carried by students aged from 8 to 10 years managed to reduce the weight of their backpacks by over 1 kg compared with a control group (mean weights of 2.28 kg after the first intervention). They also found that the results were sustained over time (mean weight of 2.24 kg at 3 months).

## 5. Conclusions

This work reinforces the role of schools as a strategic sector for improving health-related habits. The educational programmes implemented in these schools are wide in scope, and their results are perfectly extrapolated to other levels.

Study participants held incorrect postures when watching television. In addition, most did not transport school materials correctly, with the worst habits displayed by those who resided in a non-traditional family structure.

The reduction in incorrect postural hygiene habits, achieved after the intervention, will have a significant impact on future complications with respect to bones and bone structure, such as dorso-lumbar pain and muscle overload.

Encouraging good postural hygiene habits in schools and preventing bad ones are essential for health education in society in general.

## Figures and Tables

**Figure 1 healthcare-10-00864-f001:**
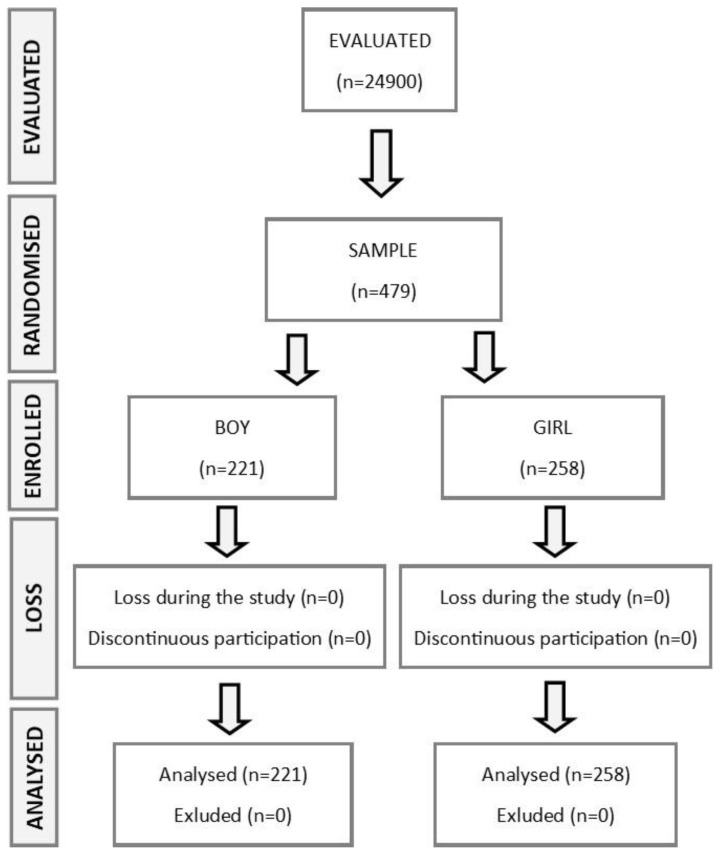
Flow diagram.

**Figure 2 healthcare-10-00864-f002:**
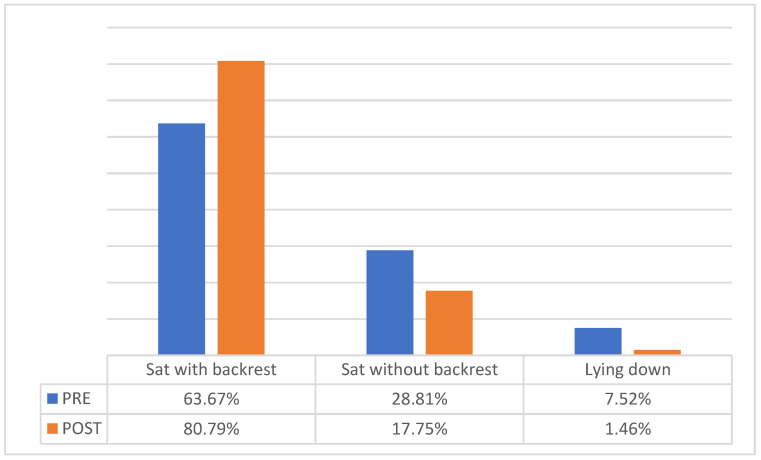
Column diagram of postural changes when watching television pre- and post-intervention.

**Figure 3 healthcare-10-00864-f003:**
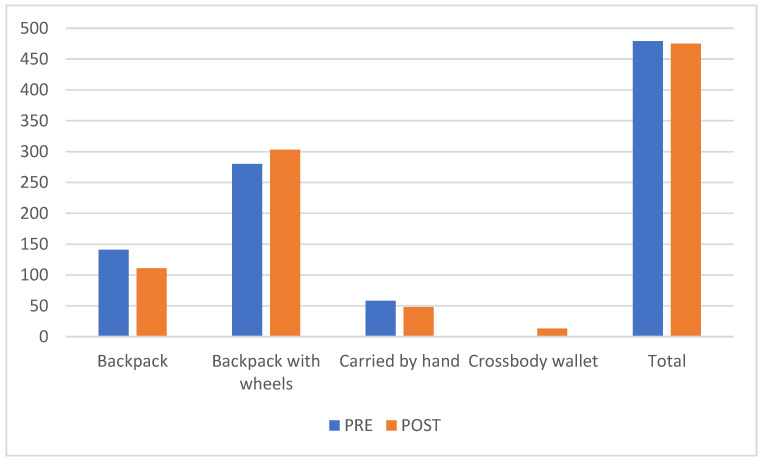
Descriptive analysis number and percentage (%) of the means of carrying school materials used by students pre- and post-intervention.

**Table 1 healthcare-10-00864-t001:** Description of the population participating in the study.

Sex	Boy	221 (46.14%)
	Girl	258 (53.86%)
Age		8.9 ± 1.49
School location	Rural	104 (21.7%)
	Urban inland	157 (32.8%)
	Urban coastal	218 (45.5%)
Number of siblings	One siblings	447 (93.3%)
	Two siblings	20 (4.2%)
	No siblings	12 (2.5%)
Members of the family unit	Parents + grandparents	101 (21.09%)
	Mother or father + grandparents	12 (2.51%)
	Grandparents	7 (1.46%)
	Mother or father	53 (11.06%)
	Mother and father	306 (63.88%)

## Data Availability

Available upon request from the corresponding author.

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
