# Peer review of "Educational Intervention in the Postural Hygiene of School-Age Children"

_healthcare, 2022, doi:10.3390/healthcare10050864_

Round 1
Reviewer 1 Report
This paper is about an intervention study for the prevention of low back pain in children. I would like to comment on the following.
L128 Do you have any references for the program PERSEO25 and the study ENKID26?
In the Material and Method section, please provide details of the educational program. There is an overview in the background, but not enough description of what was actually done in this intervention study.
L 157-158 Please describe what statistical tests were performed.
L 173-178 During the description of Fig 2, there is an abrupt description of results that are not in the figures. Please consider restructuring the paragraph.
Section 3.3. is the primary result of this study. Why not show all the results in a figure or table, not just the posture when watching TV? Also, could you perform a statistical test on the TV viewing posture?
Fig. 2 is incomprehensible. Please correct it. e.g. What means 1800.00%?
I don't understand Table 2 Footnote." What does "* * Tables may have a footer." mean?
In the Discussion, I suggest a more results-based discussion.
In Conclusion, you should include a more results-based conclusion.
L309 Why was not Institutional Review Board Statement applicable? I think the ethical review is required for intervention studies.
Please adapt the reference marking to Instruction for authors. Here is an excerpt of Instruction for authors:
"In the text, reference numbers should be placed in square brackets [ ], and placed before the punctuation; for example [1], [1-3] or [1,3]. For embedded citations in the text with pagination, use both parentheses and brackets to indicate the reference number and page number s; for example [5]. (p. 10). or [6] (pp. 101-105)."
Author Response
Dear Revisor, we hope It would be better. Thank you very much for comments and your time.

Reviewer 2 Report
This reviewer commends the researchers for their study on postural hygiene of grade school students
Introduction
- Most of the narrative in the Discussion section is more suited to the introduction section. See comment #6 below.
Method
- What is/are the research question (s). What is the research hypothesis?
- Informed consent is not sufficient to conduct human subject research. There is no evidence of ethical clearance for this study or IRB approval statement. The nature of the intervention is not clear. What exactly was the content of the health education intervention. Were the participant students of all ages in all locations provided the same volume and quality of health education intervention? Who conducted the Health education intervention? Were the intervention providers trained in health education? How long and how often was the intervention conducted? How was the quality of the intervention ensured?
Results
- on page 6, Under the subtitle, Postural Habits Post Intervention, the first sentence reads: “There was a slight decrease in the number of hours per day spent watching television 192 (p>0.001) to 2.29±0.531 hours/day. “ This does not make much sense because of absence of statistical significance. The exact p-value should be stated. The sentence should be re-articulated: “There was no decrease in the number of hours per day spent watching television 1(p=?), which was29±0.531 hours/day.”
Discussion
- The Discussion Section, on pages 8 and9: Lines 204 to 269: The narrative presented here is a review of the literature, not a discussion of the results. I suggest that this part be moved to Introduction/Background Information. The Discussion needs to be re-written.
Conclusion
The Conclusion is a broad general viewpoint and it is not centered on the specific statistically significant findings of the study. The Conclusion needs to be re-written.
Author Response

(The authors gave the same response as above.)

Round 2
Reviewer 1 Report
L101 My apologies where I did not point this out last time. This study is probably not a quasi-experimental design. It should simply be described as a pre-post intervention study.
Section 2.8. there is a lack of explanation of the statistical tests.
Are tests one-sided tests or two-sided tests?
Is the t-test paired or independent, or both? 
For paired ordinal scales, the Wilcoxon signed-rank test should be used, not the U-test.
Chi-square test and McNemar test should be used for tests of categorical variables.
Figure 2 has not been corrected. It is clearly strange.
As pointed out the previous review, Section 3.3 should show a before/after comparison of all results. This could be in a figure or table. The purpose of this study is to test whether the intervention changed behavior. It is always necessary to properly present the comparison-tested results along with the before and after data.
Please make appropriate statements in the L332-334 Institutional Review Board Statement, Informed Consent Statement, and Data Availability Statement.
Author Response
We hope it would be better, thank you for helping us improve.

Reviewer 2 Report
The first paragraph of the Discussion Section bearing the in-text citation [29]:
“WHO believes that schools play a key role in developing healthy lifestyle habits. The relationships between educational centers and health undoubtedly enhance, the correct tion of bad habits in health and work on the maintenance of new habits learned for the future. The implementation of health-related educational programs in schools is an essential way of working with a high power of dissemination and that originates favorable results of change [29].”
The above paragraph should belong in the Background section/Introduction Section. A discussion is where the results of the study are expounded using the authors' informed inferences and comparisons with similar studies in the same or different study area are made using pertinent references for such studies.
Author Response
We hope it would be better, thank you for helping us improve
